# RemedyGS: Defend 3D Gaussian Splatting against Computation Cost Attack

## Abstract

As a mainstream technique for 3D reconstruction, 3D Gaussian Splatting (3DGS) has been applied in a wide range of applications and services. Recent studies have revealed critical vulnerabilities in this pipeline and introduced computation cost attacks that lead to malicious resource occupancies and even denial-of-service (DoS) conditions, thereby hindering the reliable deployment of 3DGS. In this paper, we propose the first effective and comprehensive black-box defense framework, named *RemedyGS*, against such computation cost attacks, safeguarding 3DGS reconstruction systems and services. Our pipeline comprises two key components: a detector to identify the attacked input images with poisoned textures and a purifier to recover the benign images from their attacked counterparts, mitigating the adverse effects of these attacks. Moreover, we incorporate adversarial training into the purifier to enforce distributional alignment between the recovered and original natural images, thereby enhancing the defense efficacy. Experimental results demonstrate that our framework effectively safeguards 3DGS systems, achieving state-of-the-art performance in both safety and utility.

## 1 Introduction

3D reconstruction, which aims to synthesize photorealistic novel views from multi-view input images, plays a pivotal role in a wide range of applications, including augmented reality(AR), virtual reality(VR) (Arena et al., 2022), and holographic communication (Tu et al., 2024; Hu et al., 2024). Recently, 3D Gaussian Splatting (3DGS) (Kerbl et al., 2023) has emerged as a leading approach for 3D reconstruction. By representing scenes as a set of 3D Gaussian primitives, 3DGS enables explicit 3D modeling that significantly accelerates rendering while delivering high-quality novel view synthesis. This combination of efficiency and visual fidelity has made 3DGS attractive for commercial applications, with companies such as Spline, KIRI, and Polycam providing large-scale paid services that reconstruct 3D scenes and synthesize novel views from user-uploaded images.

The superior reconstruction capability of 3DGS stems from its adaptive density control mechanism, which introduces new Gaussian primitives to under-reconstructed areas while pruning low-contribution ones until convergence. This adaptive densification allows 3DGS to effectively capture fine geometric details and complex textures in the scene. Nevertheless, it also raises significant security concerns. Attackers can exploit this process by manipulating input images to trigger an excessive increase in the number of Gaussians, thereby significantly increasing computational costs. A recent study, Poison-splat (Lu et al., 2024a), has revealed this critical vulnerability, demonstrating how adversaries can induce dramatic escalations in GPU memory usage, training duration, and rendering latency through this new type of computation cost attacks. Instead of directly maximizing the number of Gaussians, attackers sharpen 3D objects by increasing the total variance score, which indirectly causes 3DGS to allocate more Gaussian primitives. These attacks can be launched by malicious users posing as legitimate ones or by tampering with images uploaded by others, effectively monopolizing computational resources and causing denial-of-service (DoS) conditions. Thus, the stability, reliability, and availability of real-world 3DGS systems will be severely threatened.

Several basic defense mechanisms have been proposed, such as image smoothing and limiting the number of Gaussians, which, however, are largely ineffective. Specifically, image smoothing, which employs filters such as Gaussian or bilateral filtering (Tomasi & Manduchi, 1998), aims to pre-process input images to mitigate the effects of noise introduced by attackers. Nevertheless, since

the attack process often involves complex non-linear transformations, it is ineffective to apply simple linear filters to mitigate poisoned textures. Moreover, limiting the number of Gaussians during 3DGS training may compromise the system's adaptability and representation quality, especially in complex scenes. These straightforward strategies result in an unsatisfactory trade-off between security and utility, often leading to a degradation of reconstruction quality by up to 10 dB (Lu et al., 2024a). This degradation arises from two primary reasons. First, these methods cannot differentiate between clean and poisoned images, which results in a uniform degradation in the performance of all users. Second, they fail to distinguish original textures from injected noise, thereby obscuring fine details essential for high-quality reconstructions. These limitations motivate us to explore a meticulously designed defense method that ensures the reliable and effective application of 3DGS.

In this paper, we propose RemedyGS, a comprehensive black-box defense framework to protect 3DGS systems against white-box computation cost attacks while preserving high reconstruction utility. The pipeline of RemedyGS is illustrated in Figure 1. It consists of two key components: 1) a detector that differentiates between attacked and safe input images, and 2) a learnable purifier that recovers normal images from attacked ones. Given that universal smoothing can significantly compromise the quality of reconstructions for legitimate users, we develop a detector network to identify poisoned images, ensuring that only those images flagged as compromised undergo further processing. This targeted approach preserves the utility of services for normal users while addressing the negative impacts of computation cost attacks on compromised inputs. Unlike traditional image smoothing methods, which struggle to reverse the complex transformations associated with attacked images, our learnable purifier is designed to learn the intricate non-linear inverse transformations necessary for effective recovery. This enables the purifier to achieve high-quality restoration of benign images, thereby enhancing safety under more aggressive attack scenarios. While the purifier effectively recovers images, it may produce regions that are overly blurred, leading to suboptimal reconstruction performance. To address this issue, we incorporate adversarial training to enhance the capabilities of purifier. By introducing a discriminator that provides informative feedback, we encourage alignment between the distributions of original clean images and recovered images. This adversarially trained purifier yields outputs with improved perceptual quality, facilitating more faithful 3D reconstructions. Our main contributions are summarized as follows:

- We propose the first effective black-box defense framework against computation cost attacks in 3DGS training. It comprehensively defends against existing white-box attacks, providing a system-agnostic and generic solution to safeguard 3DGS systems.

- We formulate the defense as a two-stage pipeline. A detector accurately distinguishes between attacked and safe inputs, thereby maintaining full utility for normal users, while a purifier recovers clean images from manipulated inputs, mitigating the negative impact of the attacks.

- We incorporate adversarial training into the purifier to enhance the faithful recovery of clean images, ensuring high-fidelity services for affected users. Extensive experiments demonstrate that RemedyGS provides reliable safety performance and superior utility over existing methods.

## 2 RELATED WORK

**3D Spatial Reconstruction.** Reconstructing 3D scenes from 2D visual inputs has been a long-standing problem in computer vision. This process requires transforming 2D observations into valid 3D representations. Neural Radiance Fields (NeRF) (Mildenhall et al., 2021) have significantly advanced this field by enabling high-quality novel view synthesis through neural volume rendering (Kajiya & Von Herzen, 1984), but NeRF-based methods (Müller et al., 2022; Fridovich-Keil et al., 2022; 2023) remain computationally expensive due to dense ray tracing and network inference. Recently, 3D Gaussian Splatting (3DGS) (Kerbl et al., 2023) has emerged as an efficient alternative. By representing 3D scenes explicitly with Gaussian primitives and leveraging tile-based differentiable rendering (Zwicker et al., 2001), 3DGS achieves fast optimization, high-fidelity reconstruction, and real-time rendering speed. Its adaptive Gaussian quantity control further enhances the representation capability and scalability. 3DGS further spurs research in 4D reconstruction (Sun et al., 2024; Wu et al., 2024; Li et al., 2024; Liu et al., 2024) and 3D generation (Gao et al., 2024; Chen et al., 2024; Wu et al., 2025). In this work, we focus on the recently identified vulnerabilities of 3DGS (Lu et al., 2024a) and propose an effective defense framework to ensure its safe deployment.

**Denial-of-Service Attacks and Defenses.** Denial-of-Service (DoS) attacks (Elleithy et al., 2005) traditionally exploit vulnerabilities in internet infrastructure by overwhelming servers with excessive

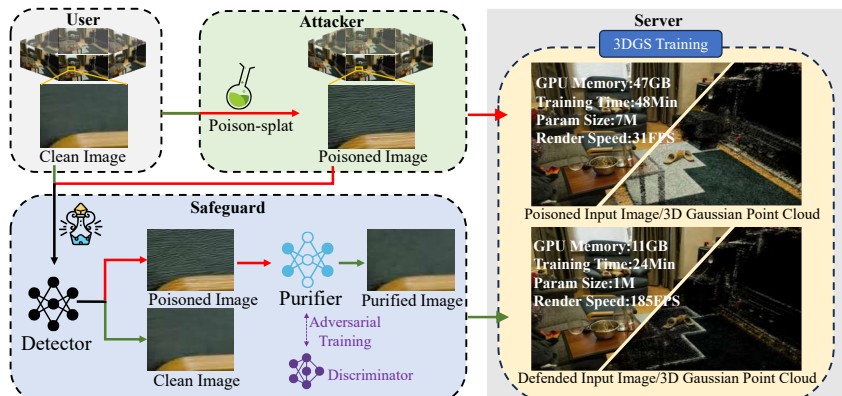

Figure 1. The overview of our proposed defense framework against 3DGS computation cost attack, where we visualize the input RGB image and 3DGS point cloud positions. The computational cost increases with the density of 3DGS point cloud. Our method effectively safeguards 3DGS systems.

requests, thereby degrading or halting services. As machine learning systems become increasingly prevalent in user-facing services, these systems have similarly become susceptible to such vulnerabilities, where adversaries can drain resources and significantly impair functionality. Previous studies have demonstrated these threats across various domains. For instance, carefully crafted inputs, such as adversarial examples (Hong et al., 2020) and backdoor triggers (Chen et al., 2023), are employed to inflate computational costs in input-adaptive networks (Hong et al., 2020) and generative models (Chen et al., 2022). In the context of 3DGS-as-a-service systems, Poison-splat (Lu et al., 2024a) represents the first exploration of DoS attacks, revealing that poisoned input can drastically increase the number of Gaussians during the densification process, thereby exhausting system resources and potentially denying service. To address these risks, existing defense strategies include game-theoretic approaches (Xu et al., 2025) and adversarial training with specialized loss functions (Wang et al., 2025). Yet, these strategies typically rely on a fixed shared model, making them incompatible with 3DGS systems that require retraining for each scene. Moreover, Poison-splat operates under a white-box setting with strong attacker assumptions, complicating the design of effective defenses. To our knowledge, no prior work has introduced an effective defense tailored for 3DGS-as-a-service systems. Our work bridges this gap by proposing a comprehensive defense framework that effectively safeguards 3DGS systems from such DoS attacks.

## 3 PRELIMINARIES

**3D Gaussian Splatting.** 3DGS (Kerbl et al., 2023) reconstructs 3D scenes from multi-view images by representing the scene as a set of learnable 3D Gaussian primitives $\mathcal{G}$. Each primitive is assigned multiple learnable properties: a spatial position vector $\boldsymbol{\mu} \in \mathbb{R}^3$, a 3D covariance matrix $\boldsymbol{\Sigma} \in \mathbb{R}^{3 \times 3}$, an opacity $o \in [0, 1]$, and a view-dependent color $\boldsymbol{c} \in \mathbb{R}^3$ modeled by spherical harmonics coefficients. It is mathematically expressed as $G(\boldsymbol{x}) = e^{-\frac{1}{2}(\boldsymbol{x}-\boldsymbol{\mu})^T \boldsymbol{\Sigma}^{-1}(\boldsymbol{x}-\boldsymbol{\mu})}$, where $\boldsymbol{x} \in \mathbb{R}^3$ represents a spatial coordinate. Using these Gaussian primitives, 3DGS renders an image by projecting the Gaussians onto the 2D image plane. The color of each pixel is computed through $\alpha$-blending the contributions of $N$ overlapping 2D Gaussian projections ordered by depth:

$$\bar{C} = \sum_{i=1}^{N} \boldsymbol{c}_i \alpha_i \prod_{j=1}^{i-1}(1 - \alpha_j), \tag{1}$$

where $\alpha_i$ denotes the transmittance of the 2D Gaussian at the $i$-th greatest depth, derived from its covariance matrix and opacity, and $\boldsymbol{c}_i$ represents its corresponding color.

The training objective of 3DGS is to minimize the difference between the rendered images, denoted by the set $\bar{\mathcal{V}} = \{\bar{\boldsymbol{V}}_k\}_{k=1}^K$ with $K$ viewpoints, and the ground-truth images, denoted by $\mathcal{V}$. This is achieved by employing a combination of $\mathcal{L}_1$ loss and the structural similarity index measure (SSIM) loss $\mathcal{L}_{\text{D-SSIM}}$, weighted by a hyperparameter $\lambda$:

$$\min_{\mathcal{G}} \mathcal{L}(\mathcal{V}, \bar{\mathcal{V}}) = (1 - \lambda)\mathcal{L}_1(\mathcal{V}, \bar{\mathcal{V}}) + \lambda \mathcal{L}_{\text{D-SSIM}}(\mathcal{V}, \bar{\mathcal{V}}). \tag{2}$$

The high-quality reconstruction capability of 3DGS relies on its density control mechanism. Instead of fixing the number of Gaussians beforehand, 3DGS employs a densification strategy that adaptively increases the number of Gaussians to compensate for under-reconstructed areas during train-

ing. Specifically, when the magnitude of the view-space positional gradient $\nabla_{\boldsymbol{\mu}} \mathcal{L}(\mathcal{V}, \bar{\mathcal{V}}) = \frac{\partial \mathcal{L}(\mathcal{V}, \bar{\mathcal{V}})}{\partial \boldsymbol{\mu}}$ exceeds a predefined threshold $\beta$, it indicates the presence of under-reconstructed areas, prompting the addition of new 3D Gaussians to enhance local detail and reconstruction fidelity.

**Poison-splat Attack.** Poison-splat (Lu et al., 2024a) is a recently proposed white-box DoS attack targeting 3DGS-as-a-service systems, where attackers masquerade as normal users by uploading poisoned images or manipulating images uploaded by others. These poisoned images disrupt the normal optimization process of 3DGS, imposing excessive computational burdens that consume substantial resources, including GPU memory, training time, and rendering latency. Consequently, this attack can push systems beyond their resource limits, leading to service unavailability and potential server breakdown. The attack is formulated as a max-min bi-level optimization problem:

$$\mathcal{V}_{\text{poi}} = \arg\max_{\mathcal{V}_{\text{poi}}} \mathcal{C}(\mathcal{G}^*) \quad \text{s.t.} \quad \mathcal{G}^* = \arg\min_{\mathcal{G}} \mathcal{L}(\mathcal{V}_{\text{poi}}), \tag{3}$$

where $\mathcal{C}$ represents the computational cost metric, $\mathcal{V}_{\text{poi}}$ denotes the poisoned image dataset, and $\mathcal{G}^*$ refers to the proxy 3DGS model trained on the poisoned dataset.

It is non-trivial to directly maximize the computational cost, as it is non-differentiable. To address this issue, Poison-splat leverages the number of Gaussians $\|\mathcal{G}\|$ as a surrogate metric, given its strong correlation with the resource consumption of 3DGS systems. The attack exploits the densification criterion $\nabla_{\boldsymbol{\mu}} \mathcal{L} > \beta$ to trigger an increase in the number of Gaussians. By introducing complex textures and non-smooth textures in the poisoned images, the positional gradient magnitude $\nabla_{\boldsymbol{\mu}} \mathcal{L}$ increases. Specifically, less smooth surfaces require more Gaussians for accurate reconstruction, thereby linking the non-smoothness of images, which is characterized by the total variance score, to the complexity of Gaussian representation, i.e., $\|\mathcal{G}\| \propto \mathcal{S}_{TV}(\mathcal{V})$. To maximize the computational cost, attackers can apply noisy perturbations to "sharpen" normal images by maximizing the total variance score of the rendered poisoned images $\mathcal{S}_{TV}(\mathcal{V}_{\text{poi}})$:

$$\mathcal{C}(\mathcal{G}) := \mathcal{S}_{TV}(\mathcal{V}_{\text{poi}}) = \sum_{\boldsymbol{V}_{\text{poi},k} \in \mathcal{V}_{\text{poi}}} \sum_{i,j} \sqrt{\left| \boldsymbol{V}_{\text{poi},k}^{i+1,j} - \boldsymbol{V}_{\text{poi},k}^{i,j} \right|^2 + \left| \boldsymbol{V}_{\text{poi},k}^{i,j+1} - \boldsymbol{V}_{\text{poi},k}^{i,j} \right|^2}. \tag{4}$$

To enhance stealth and reduce detectability for the poisoned images, the attacker enforces a perturbation constraint within an $\epsilon$-ball constraint $\mathcal{P}_\epsilon$ around the original clean images:

$$\boldsymbol{V}_{\text{poi},k} \in \mathcal{P}_\epsilon(\boldsymbol{V}_{\text{poi,k}}, \boldsymbol{V}_k) := \left\{ \boldsymbol{V}_{\text{poi},k} \big| \, \|\boldsymbol{V}_{\text{poi},k} - \boldsymbol{V}_k\|_\infty \le \epsilon \right\}, \tag{5}$$

ensuring that poisoned images remain visually similar to their clean counterparts.

Poison-splat uses a proxy 3DGS model that allows attackers to optimize Eq. (4) and Eq. (5) within a white-box 3DGS framework through exhaustive training optimizations per scene. Notably, this proxy model assumes that attackers have full access to the internal details of 3DGS systems, making naive defense methods ineffective against this attack paradigm. In contrast, our proposed RemedyGS framework leverages data-driven neural networks to effectively defend against such attacks.

## 4 REMEDYGS: A COMPREHENSIVE DEFENSE FRAMEWORK

We propose RemedyGS, a comprehensive black-box defense framework designed to protect 3DGS systems from the white-box computation cost attack. RemedyGS mitigates vulnerabilities that arise during 3DGS training, thereby enabling safe and reliable deployment of 3DGS-based reconstruction services. An overview of the framework is illustrated in Figure 1. The pipeline consists of two key components: a detector that identifies poisoned inputs and a purifier that mitigates the impact of detected attacks. Unlike prior methods that treat all input images uniformly, our framework selectively processes inputs. Specifically, a detector network first distinguishes between poisoned images $\boldsymbol{V}_{\text{poi}}$ and clean images $\boldsymbol{V}_{\text{cln}}$. Only those images flagged as poisoned are passed to the purifier. This selective approach ensures that the utility of normal users is preserved while the system remains robust against attacks. The purifier then recovers clean images $\boldsymbol{V}_{\text{rec}}$ from poisoned inputs, thereby reducing the adverse effects of the attack and ensuring stable service for legitimate users. To further enhance the fidelity of recovered images, we incorporate adversarial training. This process aligns the distribution of recovered images $\boldsymbol{V}_{\text{rec}}$ with that of original clean images $\boldsymbol{V}_{\text{cln}}$, improving perceptual quality and reconstruction accuracy. An adversarial discriminator, as illustrated in Figure 3, enforces this distributional alignment by providing informative feedback to the purifier. In summary, RemedyGS offers an effective defense against the computation cost attack without compromising high-fidelity novel view synthesis. The following subsections provide detailed descriptions of each component.

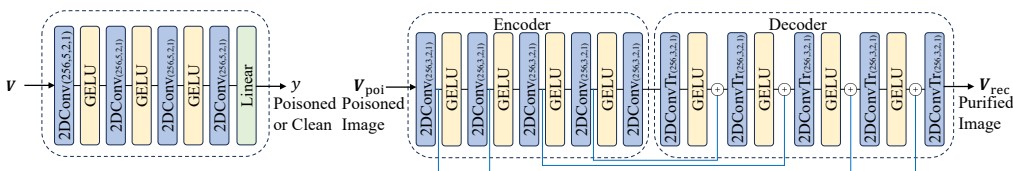

Figure 2. (Left) The architecture of our detector. (Right) The architecture of our purifier.

## 4.1 DETECTOR

The detector is a critical component in ensuring a balanced defense that preserves service quality for benign users while protecting the system against computation cost attacks. Applying purification uniformly to all inputs may impose unnecessary computational overhead and can pose unintended modifications and degrade the reconstruction quality for clean images. Therefore, it is crucial to process inputs in an adaptive and selective manner. Our detector design leverages the distinct local texture characteristics that differentiate poisoned images from clean images. This difference arises from the attack formulation, which maximizes the total variation of input images, as expressed in Eq. (4). This process introduces unnatural high-frequency noise and more pronounced edge structures to poisoned inputs. These abnormal texture patterns serve as reliable signatures for detection.

We implement the detector as a data-driven neural network $f_{\text{det}}$ trained to classify inputs as poisoned or clean. The architecture consists of four stacked 2D convolutional layers followed by a linear classification head, as illustrated in Figure 2 (Left). Convolutional kernels are particularly effective at capturing local texture features, making them well-suited to distinguish unnatural noise in poisoned images from natural textures in benign inputs. For notational simplicity, we omit the image index $k$, as the detector operates uniformly across all multi-view images of a scene. We train the network using a dataset consisting of both labeled poisoned and clean images, denoted by $\mathcal{D}_{\text{det}} = \{(\boldsymbol{V}_{\text{poi}}, y_{\text{poi}}) \cup (\boldsymbol{V}_{\text{cln}}, y_{\text{cln}})\}$, where $y_{\text{poi}} = 1$ and $y_{\text{cln}} = 0$. The loss function is expressed as:

$$\mathcal{L}_{\text{det}} = \frac{1}{|\mathcal{D}_{\text{det}}|} \sum_{i=1}^{|\mathcal{D}_{\text{det}}|} CE_{\text{loss}}(y_i, f_{\text{det}}(\boldsymbol{V}_i; \omega)), \tag{6}$$

where $|\mathcal{D}_{\text{det}}|$ denotes the size of the dataset, $\omega$ denotes the learnable parameters of the detector, $\boldsymbol{V}_i$ and $y_i$ denote the $i$-th image in the dataset and its corresponding label, respectively. Our detector identifies poisoned images characterized by unnatural textures, which are subsequently processed by the purifier, while benign images are preserved to maintain high fidelity. This targeted defense strategy maximizes reconstruction quality for normal users while safeguarding the system.

## 4.2 PURIFIER

To provide versatile 3DGS reconstruction services for normal users whose benign inputs have been manipulated and poisoned by attackers, it is essential to develop a purifier to restore these poisoned images to their original safe state. The purifier design needs to meet two requirements: (1) it should effectively remove the attacker's poison effects to avoid triggering the computation cost attack, and (2) the purified images should closely recover the original content with minimal degradation, maintaining visual consistency with users' uploads. These requirements ensure system security while preserving high-quality services for normal users. However, naive approaches such as image smoothing or restricting the number of Gaussians during reconstruction fail to meet these requirements, resulting in significant loss of detail or inadequate defense. To this end, we design a learnable purifier network based on an encoder-decoder architecture with a symmetric structure.

Given a poisoned input image $\boldsymbol{V}_{\text{poi}}$, the encoder $f_\phi$ learns to identify and eliminate the toxic textures imposed by the attackers, and progressively extracts the original image features. The decoder $g_\theta$ then reconstructs a purified image $\boldsymbol{V}_{\text{rec}} = g_\theta(f_\phi(\boldsymbol{V}_{\text{poi}}))$ from these extracted features. Suppose that the original clean images follow a distribution $p_{\text{cln}}(\boldsymbol{V}_{\text{cln}})$, while the attackers generate poisoned images through the conditional distribution $p_{\mathcal{A}}(\boldsymbol{V}_{\text{poi}}|\boldsymbol{V}_{\text{cln}})$, with the marginal distribution of poisoned images represented as $p_{\text{poi}}(\boldsymbol{V}_{\text{poi}})$. The purifier is trained to recover images $\boldsymbol{V}_{\text{rec}}$ that are as informative about the original clean images $\boldsymbol{V}_{\text{cln}}$ as possible. From an information-theoretic perspective, the training objective is formulated as maximizing the mutual information between $\boldsymbol{V}_{\text{rec}}$ and $\boldsymbol{V}_{\text{cln}}$, i.e., $\max_{\phi,\theta} I(\boldsymbol{V}_{\text{cln}}; \boldsymbol{V}_{\text{rec}})$. Since directly solving this optimization problem is intractable, we employ the Barber–Agakov bound, as presented in Theorem 1, to obtain a tractable surrogate objective.

**Theorem 1** (Barber-Agakov Bound (Barber & Agakov, 2004)). *Let $x$ and $y$ be random variables. The mutual information between them admits the following variational lower bound:*

$$I(x; y) \geq H(x) + \mathbb{E}_{p(x,y)}\big[\log q(x|y)\big], \tag{7}$$

*where $q(x|y)$ is an arbitrary variational distribution.*

Applying this bound to our problem, we derive:

$$I(\boldsymbol{V}_{\text{cln}}; \boldsymbol{V}_{\text{rec}}) \geq H(\boldsymbol{V}_{\text{cln}}) + \mathbb{E}_{p(\boldsymbol{V}_{\text{cln}}, \boldsymbol{V}_{\text{rec}})}\left[\log q(\boldsymbol{V}_{\text{cln}}|\boldsymbol{V}_{\text{rec}})\right]. \tag{8}$$

Moreover, we assume that $q(\boldsymbol{V}_{\text{cln}}|\boldsymbol{V}_{\text{rec}})$ follows an independent multivariate Gaussian distribution, i.e., $q(\boldsymbol{V}_{\text{cln}}|\boldsymbol{V}_{\text{rec}}) \sim \mathcal{N}(\boldsymbol{V}_{\text{rec}}, \sigma^2 \boldsymbol{I})$, where $\sigma^2$ denotes the variance of the Gaussian distribution and $\boldsymbol{I}$ is the identity matrix. This assumption facilitates further simplification:

$$I(\boldsymbol{V}_{\text{cln}}; \boldsymbol{V}_{\text{rec}}) \geq H(\boldsymbol{V}_{\text{cln}}) + \log \frac{1}{2\pi\sigma^d} - \mathbb{E}_{p(\boldsymbol{V}_{\text{cln}}, \boldsymbol{V}_{\text{rec}})} \frac{\|\boldsymbol{V}_{\text{cln}} - \boldsymbol{V}_{\text{rec}}\|^2}{2\sigma^2}, \tag{9}$$

where $d$ denotes the dimension of $\boldsymbol{V}_{\text{cln}}$. Notably, $H(\boldsymbol{V}_{\text{cln}})$ and $\sigma$ are constants, allowing us to ignore them in the optimization. Therefore, minimizing the term $\mathbb{E}_{p(\boldsymbol{V}_{\text{cln}}, \boldsymbol{V}_{\text{rec}})} \frac{\|\boldsymbol{V}_{\text{cln}} - \boldsymbol{V}_{\text{rec}}\|^2}{2\sigma^2}$ maximizes the lower bound of $I(\boldsymbol{V}_{\text{cln}}; \boldsymbol{V}_{\text{rec}})$, and, consequently, maximizes the mutual information. Thus, the training objective can be formulated as:

$$\mathcal{L}_{\text{pur}} = \min_{\phi, \theta} \mathbb{E}_{p_{\text{cln}}(\boldsymbol{V}_{\text{cln}})} \mathbb{E}_{p_{\mathcal{A}}(\boldsymbol{V}_{\text{poi}}|\boldsymbol{V}_{\text{cln}})} \|\boldsymbol{V}_{\text{cln}} - g_\theta(f_\phi(\boldsymbol{V}_{\text{poi}}))\|_2^2. \tag{10}$$

This provides a practical and effective training criterion for the purifier.

As shown in the structure of the purifier network in Figure 2 (Right), our encoder comprises a stack of five 2D convolutional layers, while the decoder symmetrically employs transposed convolution layers. To preserve finer details of the original images, skip connections are utilized between the encoder and decoder, highlighted as blue lines in Figure 2 (Right). Unlike naive image smoothing methods, our purifier learns a complex non-linear transformation that maps the poisoned image distribution $p_{\text{poi}}(\boldsymbol{V}_{\text{poi}})$ to a purified distribution $p_{\text{rec}}(\boldsymbol{V}_{\text{rec}})$, enabling high-fidelity image recovery and achieving superior safety performance against stronger attacks.

### 4.3 ADVERSARIAL TRAINING

While the purifier introduced in the previous subsection effectively defends against the computation cost attack and recovers benign images from poisoned inputs, we observe that the recovered images often contain overly blurred regions, as illustrated in Figure 6. This phenomenon arises from a common limitation of convolutional neural networks trained with the mean squared error (MSE) loss, which tends to produce overly smooth outputs. Such smoothing compromises the fine details and perceptual quality, adversely impacting the user experience and reconstruction performance.

To address this issue and better align the purified images with the original clean distribution, we incorporate adversarial training into the purifier design. In this framework, the purifier is optimized under the supervision of a discriminator network $\mathcal{F}$, which enforces distributional consistency between the purified images and the natural clean images. Specifically, the discriminator is trained to distinguish between images sampled from distribution $p_{\text{cln}}(\boldsymbol{V}_{\text{cln}})$ and those drawn from $p_{\text{rec}}(\boldsymbol{V}_{\text{rec}})$, providing informative supervision that encourages the purifier to generate outputs that are more faithfully aligned with the original distribution $p_{\text{cln}}(\boldsymbol{V}_{\text{cln}})$. To enhance the capability of the discriminator, we introduce auxiliary conditioning information derived from the latent representations of both clean and purified images, which are extracted by the purifier's encoder. This auxiliary information, denoted by $\boldsymbol{c}$, enriches the discriminator's input and improves its ability to learn subtle differences. The training objective for the discriminator $\mathcal{F}$ is formulated as follows:

$$\mathcal{L}_{\mathcal{F}} = \min_{\mathcal{F}} -\mathbb{E}_{p_{\text{cln}}} \log(\mathcal{F}(\boldsymbol{V}|\boldsymbol{c})) - \mathbb{E}_{p_{\text{rec}}} \log(1 - \mathcal{F}(\boldsymbol{V}|\boldsymbol{c})), \tag{11}$$

where $\mathcal{F}(\boldsymbol{V}|\boldsymbol{c})$ outputs the probability that the input image $\boldsymbol{V}$ comes from the clean distribution given condition $\boldsymbol{c}$. Given a fixed purifier, the optimal discriminator $\mathcal{F}^*$ satisfies:

$$\mathcal{F}^*(\boldsymbol{V}|\boldsymbol{c}) = \frac{p_{\text{cln}}(\boldsymbol{V})}{p_{\text{cln}}(\boldsymbol{V}) + p_{\text{rec}}(\boldsymbol{V})}, \tag{12}$$

with derivation provided in Appendix A. With this optimal discriminator, the purifier is further enhanced, being enforced to generate samples from $p_{\text{rec}}(\boldsymbol{V}_{\text{rec}})$ that are indistinguishable from those drawn from the natural image distribution $p_{\text{cln}}(\boldsymbol{V}_{\text{cln}})$. This adversarial training effectively promotes

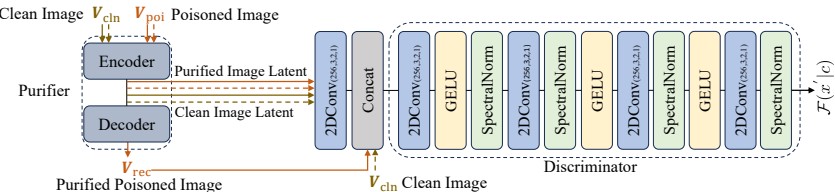

Figure 3. The architecture of our adversarial training framework.

alignment between the two distributions through an interactive process, as formulated below:

$$\mathcal{L}_G = \max_{\phi,\theta} \left\{ -\mathbb{E}_{p_{\text{cln}}} \log(\mathcal{F}^*(\boldsymbol{V}|\boldsymbol{c})) - \mathbb{E}_{p_{\text{rec}}} \log(1 - \mathcal{F}^*(\boldsymbol{V}|\boldsymbol{c})) \right\}. \tag{13}$$

This objective is maximized if and only if the reconstructed distribution matches the clean distribution, i.e., $p_{\text{rec}} = p_{\text{cln}}$. The derivation of this conclusion is presented in Appendix B, indicating that incorporating a discriminator theoretically achieves faithful alignment between $p_{\text{rec}}$ and $p_{\text{cln}}$.

Our adversarial training architecture is illustrated in Figure 3. Specifically, we construct a discriminator using a stack of four 2D convolution layers. The condition latents are first upscaled via a 2D convolution to match the spatial dimensions of the input image, then concatenated along the channel dimension with either purified or clean images before being fed into the discriminator, which predicts whether the concatenated images are drawn from $p_{\text{cln}}$ or $p_{\text{rec}}$.

Within the adversarial training framework, the purifier and the discriminator are updated alternately. The purifier is optimized to fool the discriminator, improving its ability to remove poisoned textures and enhance recovery quality. Following this, the discriminator is updated to better distinguish between purified and clean images, allowing it to guide the next evolution of the purifier. After several iterations of alternating training, the purifier can recover images that retain the original textures with minimal degradation. In practice, we use the learned perceptual image patch similarity (LPIPS) loss to aid in training. The training objective for the purifier is formulated as a weighted combination of the LPIPS loss, the MSE loss, and the adversarial training loss $\mathcal{L}_G$, expressed as:

$$\mathcal{L}'_{\text{pur}} = \alpha_1 \mathcal{L}_{\text{MSE}} + \alpha_2 \mathcal{L}_{\text{LPIPS}} + \alpha_3 \mathcal{L}_G, \tag{14}$$

where $\alpha_1$, $\alpha_2$, and $\alpha_3$ denote the weights for the MSE loss, LPIPS loss, and $\mathcal{L}_G$, respectively. Meanwhile, the discriminator is trained using $\mathcal{L}_{\mathcal{F}}$ in Eq. (11).

## 5 EXPERIMENTS

In this section, we evaluate our defense framework against white-box computation cost attacks. Due to space limitations, the evaluation in the black-box setting is deferred to Appendix D.1.

### 5.1 EXPERIMENTAL SETUP

**Baselines.** We utilize the official implementation of Poison-splat (Lu et al., 2024a) in the white-box setting to simulate attacker behavior and the official implementation of vanilla 3DGS (Kerbl et al., 2023) to simulate victim behavior. We compare our RemedyGS with two baseline defense methods from (Lu et al., 2024a): *image smoothing* and *limiting the number of Gaussians*. The former pre-processes all server-received images with a Gaussian filter to remove high-frequency components and mitigate noise introduced by attackers, while the latter imposes an upper bound on the number of Gaussians during training, thereby controlling the computational cost in a straightforward manner.

**Datasets.** To train the detector and purifier networks of RemedyGS, we curate a dataset consisting of paired benign and poisoned images. This dataset is based on the DL3DV dataset (Ling et al., 2024), which provides large-scale multi-view calibrated images at a resolution of $960 \times 540$. We sample 320 scenes and apply the computation cost attack on each, yielding a total of 1 million pairs of clean and poisoned images for training. For evaluation, we adopt three popular 3D reconstruction benchmark datasets: NeRF-Synthetic (Mildenhall et al., 2021) with 8 synthetic objects, Mip-NeRF360 (Barron et al., 2022) with 9 scenes containing complex central areas and detailed backgrounds, and Tanks-and-Temples (Knapitsch et al., 2017) with 21 real-world outdoor and indoor scenes.

**Evaluation Metrics.** An optimal defense method must balance safety and utility. In the context of computation cost attacks, the safety performance is primarily measured by GPU memory usage, with auxiliary indicators including the number of 3D Gaussians, training time, and rendering speed.

Table 1. Comparison of baselines with RemedyGS on poisoned data, evaluating computational costs like the number of Gaussians and peak GPU memory usage across three benchmark datasets.

| Metric | Number of Gaussians | | | | | Peak GPU memory [MB] | | | | |
| Setting
Scene | clean
(Ground truth) | poisoned | image
smoothing | limiting
Gaussian
number | RemedyGS
(Ours) | clean
(Ground truth) | poisoned | image
smoothing | limiting
Gaussian
number | RemedyGS
(Ours) |
|---|---|---|---|---|---|---|---|---|---|---|
| NS-chair | 0.495 M | 0.942 M | 0.163 M | 0.508 M | 0.491 M | 4633 | 9378 | 4066 | 4679 | 4912 |
| NS-ficus | 0.267 M | 0.288 M | 0.122 M | 0.289 M | 0.217 M | 4052 | 5447 | 3935 | 5418 | 4034 |
| NS-Avg | 0.289 M | 0.709 M | 0.130 M | 0.448 M | 0.327 M | 4308 | 11259 | 4027 | 5597 | 4601 |
| MIP-bonsai | 1.275 M | 6.194 M | 0.838 M | 2.095 M | 1.034 M | 9066 | 19476 | 8459 | 10709 | 8663 |
| MIP-room | 1.536 M | 7.368 M | 0.847 M | 2.092 M | 1.178 M | 11370 | 47721 | 9976 | 12266 | 10965 |
| MIP-Avg | 3.179 M | 7.037 M | 1.700 M | 3.766 M | 2.496 M | 12136 | 23961 | 9083 | 13237 | 10630 |
| TT-Barn | 0.999 M | 1.848 M | 0.396 M | 1.018 M | 0.589 M | 6063 | 8352 | 4638 | 5962 | 5128 |
| TT-Francis | 0.766 M | 1.589 M | 0.283 M | 1.030 M | 0.429 M | 4130 | 5805 | 3580 | 4644 | 4012 |
| TT-Avg | 1.750 M | 2.874 M | 0.728 M | 1.968 M | 1.075 M | 7052 | 9608 | 4881 | 7753 | 5608 |

Table 2. Comparison of baselines with RemedyGS on poisoned data, evaluated by utility metrics including PSNR, LPIPS, and SSIM across three benchmark datasets.

| Metric | PSNR ↑ | | | | | LPIPS ↓ | | | | | SSIM ↑ | | | | |
| Setting
Scene | clean
(Ground truth) | poisoned | image
smoothing | limiting
Gaussian
number | RemedyGS
(Ours) | clean
(Ground truth) | poisoned | image
smoothing | limiting
Gaussian
number | RemedyGS
(Ours) | clean
(Ground truth) | poisoned | image
smoothing | limiting
Gaussian
number | RemedyGS
(Ours) |
|---|---|---|---|---|---|---|---|---|---|---|---|---|---|---|---|
| NS-chair | 35.776 | 32.133 | 29.635 | 32.005 | **34.261** | 0.010 | 0.055 | 0.064 | 0.053 | **0.031** | 0.988 | 0.943 | 0.941 | 0.945 | **0.9731** |
| NS-ficus | 35.542 | 33.085 | 31.407 | 33.105 | **35.483** | 0.012 | 0.040 | 0.037 | 0.040 | **0.016** | 0.987 | 0.968 | 0.963 | 0.968 | **0.984** |
| NS-Avg | 33.866 | 30.785 | 30.029 | 30.767 | **33.070** | 0.030 | 0.094 | 0.076 | 0.091 | **0.057** | 0.969 | 0.905 | 0.938 | 0.913 | **0.955** |
| MIP-bonsai | 32.258 | 27.001 | 29.989 | 24.974 | **31.320** | 0.184 | 0.476 | 0.255 | 0.432 | **0.228** | 0.947 | 0.636 | 0.878 | 0.747 | **0.924** |
| MIP-room | 31.717 | 26.458 | 29.901 | 23.238 | **31.112** | 0.199 | 0.472 | 0.286 | 0.442 | **0.248** | 0.927 | 0.550 | 0.844 | 0.653 | **0.899** |
| MIP-Avg | 27.520 | 24.704 | 26.446 | 22.570 | **27.310** | 0.222 | 0.417 | 0.321 | 0.442 | **0.265** | 0.813 | 0.592 | 0.736 | 0.607 | **0.792** |
| TT-Barn | 28.496 | 25.298 | 26.140 | 25.105 | **27.439** | 0.182 | 0.355 | 0.315 | 0.348 | **0.245** | 0.869 | 0.643 | 0.770 | 0.697 | **0.826** |
| TT-Francis | 28.181 | 25.725 | 26.551 | 24.146 | **27.566** | 0.240 | 0.415 | 0.343 | 0.393 | **0.287** | 0.910 | 0.724 | 0.844 | 0.770 | **0.882** |
| TT-Avg | 24.256 | 22.937 | 23.284 | 22.098 | **24.073** | 0.194 | 0.374 | 0.319 | 0.377 | **0.254** | 0.844 | 0.645 | 0.765 | 0.674 | **0.816** |

A defense is considered strong if the computational cost approaches that of the benign case. The utility is evaluated based on the 3D reconstruction quality using peak signal-to-noise ratio (PSNR), structural similarity index measure (SSIM), and learned perceptual image patch similarity (LPIPS).

## 5.2 MAIN RESULTS

We evaluate RemedyGS against the Poison-splat computation cost attack. The quantitative results for safety and utility across the three benchmark datasets are summarized in Tables 1 and 2, with additional metrics on training time and rendering speed provided in Appendix D.3. The results demonstrate that RemedyGS achieves state-of-the-art performance in both safety and utility. As shown in Table 1, the poisoned images generated by computation cost attacks trigger a significantly increased number of Gaussians, leading to more than $2\times$ GPU memory usage compared with benign cases. Although image smoothing can alleviate this by reducing high-frequency image components, it leads to over-defense issues, causing up to 9 dB degradation in reconstruction quality due to the corruption of the input images fed into 3DGS, which significantly impacts the user service utility. Similarly, limiting the Gaussian quantity through an upper bound controls the computational cost but results in notable utility loss, as sharpened regions force reallocation of Gaussians from unsharpened areas (see visualizations in Appendix D.4). In contrast, our RemedyGS effectively mitigates the attack, maintaining computational costs comparable to benign cases while preserving high-quality 3D representations. Specifically, compared to naive defense baselines, RemedyGS improves PSNR by up to 4 dB and increases SSIM by 0.24. In addition, our method aligns the defense results with natural image distributions, demonstrating significant improvements in the LPIPS score. Qualitative results are presented in Figure 4, where our RemedyGS demonstrates superior representation performance with high-fidelity recovery and consistent visual textures. These significant performance gains stem from two key designs. First, our purifier learns to perform complex non-linear transformations from poisoned data to original representations, ensuring faithful recovery while mitigating the risk of the attack. Second, our adversarial training strategy enforces the alignment between purified and clean image distributions, enhancing perceptual quality and reconstruction fidelity. In conclusion, our RemedyGS achieves the best utility performance while safeguarding the systems.

To evaluate our detector's capability in distinguishing poisoned images from clean images, we test it on a mixed dataset of clean and poisoned images across the three benchmark datasets. The results in Table 3 show that our detector effectively identifies poisoned images. Moreover, to validate the effectiveness of our detector in preserving the utility for normal users, we evaluate the

Table 3. Results of the detector.

| Metric
Scene | Accuracy | F1 | Recall |
|---|---|---|---|
| NS-all | 0.9737 | 0.9738 | 0.9737 |
| MIP-all | 0.9936 | 0.9936 | 0.9936 |
| TT-all | 0.9400 | 0.9400 | 0.9400 |

defense methods on clean images in Table 4. Unlike image smoothing that treat all inputs uniformly, our detector effectively identifies the majority of clean inputs and bypasses unnecessary processing. Consequently, our method achieves utility performance nearly identical to the original 3DGS for normal users. Evaluation results of the robustness of our method under varying attack strengths are provided in Appendix D.2.

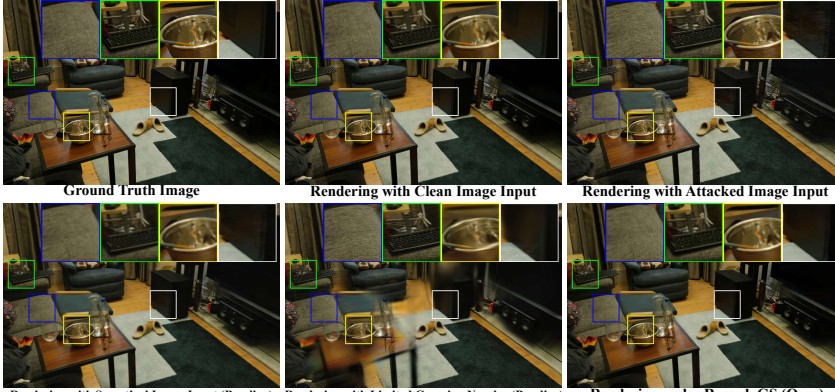

**Figure 4.** Qualitative results of rendered images for the room scene in the Mip-NeRF360 dataset. Top row: (left) ground truth image, (middle) rendering with clean image input, (right) rendering with attacked image input. Bottom row: (left) rendering with smoothed image input, (middle) rendering with an upper bound on the number of Gaussians, (right) rendering under our RemedyGS.

**Table 4.** Comparison between the baseline *image smoothing* with a uniform defense paradigm and our RemedyGS with a specialized detector, evaluated on clean data.

| Metric | PSNR ↑ | | | LPIPS ↓ | | | SSIM ↑ | | |
|---|---|---|---|---|---|---|---|---|---|
| Setting
Scene | clean
(Ground truth) | image
smoothing | RemedyGS
(Ours) | clean
(Ground truth) | image
smoothing | RemedyGS
(Ours) | clean
(Ground truth) | image
smoothing | RemedyGS
(Ours) |
| MIP-bonsai | 32.258 | 31.974 | **32.258** | 0.184 | 0.195 | **0.184** | 0.947 | 0.941 | **0.947** |
| MIP-kitchen | 31.520 | 31.014 | **31.578** | 0.117 | 0.137 | **0.117** | 0.933 | 0.924 | **0.933** |
| NS-chair | 35.776 | 27.078 | **35.776** | 0.010 | 0.070 | **0.010** | 0.988 | 0.925 | **0.988** |
| NS-ficus | 35.542 | 22.926 | **34.472** | 0.012 | 0.074 | **0.013** | 0.987 | 0.893 | **0.983** |

**Table 5.** Ablation study on different components of our RemedyGS framework.

| Metric | PSNR ↑ | | | LPIPS ↓ | | | SSIM↑ | | |
|---|---|---|---|---|---|---|---|---|---|
| Scene
Method | NS-chair | NS-ficus | MIP-room | NS-chair | NS-ficus | MIP-room | NS-chair | NS-ficus | MIP-room |
| CNN | 30.964 | 32.995 | 30.577 | 0.057 | 0.028 | 0.287 | 0.950 | 0.974 | 0.886 |
| + Concatenate | 31.331 | 33.982 | 30.418 | 0.047 | 0.022 | 0.264 | 0.959 | 0.979 | 0.887 |
| + Add | 33.868 | 35.028 | 30.997 | 0.033 | 0.019 | 0.261 | **0.973** | 0.983 | **0.899** |
| + Add + Adv. | **34.261** | **35.483** | **31.112** | **0.031** | **0.016** | **0.248** | **0.973** | 0984 | **0.899** |

## 5.3 ABLATION STUDY

We conduct an ablation study to evaluate the contributions of different components in RemedyGS. We compare several variants of the purifier and our proposed design in Table 5. Compared with the naive architecture without skip connections (CNN), our design with additive skip connections achieves consistently higher utility across all metrics and datasets. This improvement stems from the ability to extract benign patterns through the encoder and inject these fine-grained features into the decoder, thereby preserving high fidelity. We also compare our additive skip connections with channel-wise concatenation. The channel-wise concatenation approach (+ Concatenate) fails to provide high utility performance, since the concatenation operation propagates attacker-induced noise along with useful features. In contrast, additive skip connections mitigate this effect, enhancing the preservation of original details and improving overall utility. Compared with the variant without adversarial training, our proposed purifier achieves lower LPIPS scores, indicating superior perceptual quality. The effectiveness of adversarial training is further illustrated in Appendix D.4.

## 6 CONCLUSION

In this paper, we propose RemedyGS, the first comprehensive black-box defense framework against computation cost attacks in 3DGS systems. This framework facilitates the reliable and efficient deployment of 3DGS for real-world novel view synthesis applications. RemedyGS integrates a detector that effectively discriminates between adversarially manipulated and benign input images with a purifier that restores clean images from their poisoned counterparts. To further enhance the quality of the recovered images, we incorporate adversarial training into the purifier, ensuring that the purified outputs closely align with the original clean data distribution. Extensive experiments demonstrate that RemedyGS achieves state-of-the-art performance in both system safety and reconstruction utility, thereby enabling secure and high-fidelity novel view synthesis services.

REPRODUCIBILITY STATEMENT

To ensure community-wide reproducibility, we provide detailed descriptions of the experimental setup, model configurations, and evaluation procedures in the main paper and Appendix. We will also make the source code publicly available upon publication, thereby enabling researchers to independently verify and reliably replicate our results.

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

# Appendix

## A DERIVATION FOR OPTIMAL DISCRIMINATOR

In this section, we rigorously derive the expression of the optimal discriminator in Eq. (12). Note that

$$\mathcal{L}_{\mathcal{F}} = \min_{\mathcal{F}} -\mathbb{E}_{p_{\text{cln}}} \log(\mathcal{F}(\boldsymbol{V}|\boldsymbol{c})) - \mathbb{E}_{p_{\text{rec}}} \log(1 - \mathcal{F}(\boldsymbol{V}|\boldsymbol{c})) \tag{15}$$

$$= \min_{\mathcal{F}} \int_{\boldsymbol{V}} -p_{cln}(\boldsymbol{V}) \log(\mathcal{F}(\boldsymbol{V}|c)) - p_{rec}(\boldsymbol{V}) \log(1 - \mathcal{F}(\boldsymbol{V}|c)) \mathrm{d}\boldsymbol{V}. \tag{16}$$

For any $\alpha \in \mathbb{R}\setminus\{0\}$ and $\beta \in \mathbb{R}\setminus\{0\}$, considering a function $f(x) = \alpha \log(x) + \beta \log(1-x)$ with $x \in [0,1]$, $f(x)$ achieves its maximum at $x = \frac{\alpha}{\alpha+\beta}$. Therefore, given a fixed purifier, the optimal discriminator $\mathcal{F}^*$ is given by

$$\mathcal{F}^* = \frac{p_{\text{cln}}(\boldsymbol{V})}{p_{\text{cln}}(\boldsymbol{V}) + p_{\text{rec}}(\boldsymbol{V})}. \tag{17}$$

## B DERIVATION FOR THE MAXIMUM ADVERSARIAL TRAINING OBJECTIVE

In this section, we provide the detailed derivation of the conclusion that Eq. (13) is maximized if and only if the reconstructed distribution matches the clean distribution (Goodfellow et al., 2020). Specifically, we have

$$\mathcal{L}_G = \max_{\phi,\theta} \left\{ -\mathbb{E}_{p_{\text{cln}}} \log(\mathcal{F}^*(\boldsymbol{V}|\boldsymbol{c})) - \mathbb{E}_{p_{\text{rec}}} \log(1 - \mathcal{F}^*(\boldsymbol{V}|\boldsymbol{c})) \right\} \tag{18}$$

$$= \max_{\phi,\theta} \left\{ -\mathbb{E}_{p_{\text{cln}}} \log\left( \frac{p_{\text{cln}}(\boldsymbol{V})}{p_{\text{cln}}(\boldsymbol{V}) + p_{\text{rec}}(\boldsymbol{V})} \right) - \mathbb{E}_{p_{\text{rec}}} \log\left( \frac{p_{\text{rec}}(\boldsymbol{V})}{p_{\text{cln}}(\boldsymbol{V}) + p_{\text{rec}}(\boldsymbol{V})} \right) \right\} \tag{19}$$

$$= \max_{\phi,\theta} \left\{ -\mathbb{E}_{p_{\text{cln}}} \log\left( \frac{p_{\text{cln}}(\boldsymbol{V})}{2 \cdot \frac{p_{\text{cln}}(\boldsymbol{V}) + p_{\text{rec}}(\boldsymbol{V})}{2}} \right) - \mathbb{E}_{p_{\text{rec}}} \log\left( \frac{p_{\text{rec}}(\boldsymbol{V})}{2 \cdot \frac{p_{\text{cln}}(\boldsymbol{V}) + p_{\text{rec}}(\boldsymbol{V})}{2}} \right) \right\} \tag{20}$$

$$= \max_{\phi,\theta} \left\{ 2\log 2 - \mathbb{E}_{p_{\text{cln}}} \log\left( \frac{p_{\text{cln}}(\boldsymbol{V})}{\frac{p_{\text{cln}}(\boldsymbol{V}) + p_{\text{rec}}(\boldsymbol{V})}{2}} \right) - \mathbb{E}_{p_{\text{rec}}} \log\left( \frac{p_{\text{rec}}(\boldsymbol{V})}{\frac{p_{\text{cln}}(\boldsymbol{V}) + p_{\text{rec}}(\boldsymbol{V})}{2}} \right) \right\} \tag{21}$$

$$= \max_{\phi,\theta} \left\{ 2\log 2 - KL\left( p_{\text{cln}}(\boldsymbol{V}) || \frac{p_{\text{cln}}(\boldsymbol{V}) + p_{\text{rec}}(\boldsymbol{V})}{2} \right) - KL\left( p_{\text{rec}}(\boldsymbol{V}) || \frac{p_{\text{cln}}(\boldsymbol{V}) + p_{\text{rec}}(\boldsymbol{V})}{2} \right) \right\}, \tag{22}$$

where $KL(p_{\text{cln}}(\boldsymbol{V}) || \frac{p_{\text{cln}}(\boldsymbol{V}) + p_{\text{rec}}(\boldsymbol{V})}{2})$ and $KL(p_{\text{rec}}(\boldsymbol{V}) || \frac{p_{\text{cln}}(\boldsymbol{V}) + p_{\text{rec}}(\boldsymbol{V})}{2})$ denote the Kullback–Leibler divergence. According to the definition of the Jensen-Shannon divergence, we have

$$\mathcal{L}_G = \max_{\phi,\theta} 2\log 2 - JSD(p_{\text{cln}}(\boldsymbol{V}) || p_{\text{rec}}(\boldsymbol{V})), \tag{23}$$

where $JSD$ denotes the Jensen-Shannon divergence. Since $JSD$ is nonnegative, $\mathcal{L}_G$ achieves its maximum when $JSD(p_{\text{cln}}(\boldsymbol{V}) || p_{\text{rec}}(\boldsymbol{V})) = 0$, which occurs if and only if $p_{\text{rec}}(\boldsymbol{V}) = p_{\text{cln}}(\boldsymbol{V})$. Therefore, $\mathcal{L}_G$ is maximized precisely when the reconstructed distribution matches the clean distribution.

## C IMPLEMENTATION DETAILS

**Detector.** We train the detector on a curated dataset that consists of paired benign and poisoned images. During detector training, the images are resized to $960 \times 528$. The detector is trained with a learning rate of $1 \times 10^{-4}$ for 50 epochs using a batch size of 16. During the evaluation, images from the Mip-NeRF360 dataset with a resolution of $1600 \times 1066$ are randomly cropped to the same resolution as the training data. Due to the locally injected poisoned textures, the detector can effectively identify poisoned images based on these randomly cropped samples.

**Purifier.** We also train the purifier on the curated dataset that consists of paired benign and poisoned images. During training, we resize all input images to a fixed resolution of $960 \times 544$ to guarantee

Table 6. Comparison of the *image smoothing* baseline with our RemedyGS framework against black-box attacks, evaluated in terms of computational cost metrics including the number of Gaussians, peak GPU memory usage, training time, and rendering speed on the Mip-NeRF360 dataset. Compared with the baseline, our RemedyGS demonstrates the strongest capability to defend against computation cost attacks.

| Metric | Number of Gaussians | | | Peak GPU memory [MB] | | | Training time [minutes] | | | Rendering speed [FPS] | | |
|---|---|---|---|---|---|---|---|---|---|---|---|---|
| Setting / Scene | clean | image smoothing | RemedyGS (Ours) | clean | image smoothing | RemedyGS (Ours) | clean | image smoothing | RemedyGS (Ours) | clean | image smoothing | RemedyGS (Ours) |
| MIP-bonsai | 4.396 M | 3.833 M | 3.947 M | 9454 | 8685 | 8944 | 25.70 | 28.15 | 24.47 | 172 | 94 | 184 |
| MIP-counter | 2.843 M | 2.439 M | 2.584 M | 10586 | 8814 | 9106 | 30.42 | 31.63 | 30.68 | 41 | 43 | 44 |
| MIP-kitchen | 3.395 M | 3.125 M | 3.190 M | 11320 | 10513 | 10116 | 31.56 | 34.27 | 29.92 | 42 | 42 | 38 |
| MIP-room | 2.924 M | 2.229 M | 2.454 M | 11823 | 10097 | 10655 | 28.92 | 29.51 | 25.04 | 46 | 49 | 39 |

Table 7. Comparison of the *image smoothing* baseline with our RemedyGS framework against black-box attacks, evaluated in terms of utility metrics including PSNR, LPIPS, and SSIM on the Mip-NeRF360 dataset. Our RemedyGS demonstrates superior performance by effectively recovering clean images from poisoned inputs manipulated by black-box attacks.

| Metric | PSNR ↑ | | | LPIPS ↓ | | | SSIM ↑ | | |
|---|---|---|---|---|---|---|---|---|---|
| Setting / Scene | clean (Ground truth) | image smoothing | RemedyGS (Ours) | clean (Ground truth) | image smoothing | RemedyGS (Ours) | clean (Ground truth) | image smoothing | RemedyGS (Ours) |
| MIP-bonsai | 32.884 | 30.289 | **31.554** | 0.182 | 0.253 | **0.228** | 0.948 | 0.883 | **0.925** |
| MIP-counter | 29.479 | 28.318 | **28.926** | 0.184 | 0.266 | **0.233** | 0.917 | 0.853 | **0.884** |
| MIP-kitchen | 31.426 | 29.268 | **30.607** | 0.119 | 0.216 | **0.158** | 0.932 | 0.861 | **0.899** |
| MIP-room | 32.085 | 30.167 | **31.311** | 0.190 | 0.281 | **0.243** | 0.931 | 0.847 | **0.903** |

that the input images can be faithfully reconstructed with consistent spatial resolution after passing through the encoder–decoder pipeline. The purifier is trained with a learning rate of $1 \times 10^{-4}$ for 50 epochs using a batch size of 16. During the evaluation, images from the Mip-NeRF360 dataset are resized to $1600 \times 1056$.

**Adversarial Training.** For adversarial training of the purifier, we first employ a warm-up model trained solely with the MSE loss in Eq. (10) to stabilize the optimization. We then fix the parameters of this warm-up purifier and train a discriminator to distinguish between the original clean images and the outputs produced by the warm-up purifier. During this stage, the discriminator is trained with a learning rate of $5 \times 10^{-4}$ for 50 epochs. The discriminator and the warm-up purifier are subsequently used to initialize the adversarial training process, during which the discriminator and purifier are alternately updated. Specifically, the purifier is optimized with a weighted combination of three losses, as defined in Eq. 14, with weights set to $\alpha_1 = 0.23$, $\alpha_2 = 100$, and $\alpha_3 = 1$, respectively. The learning rates for both the purifier and the discriminator are set to $2.5 \times 10^{-4}$ and the epoch is set to 40.

# D  ADDITIONAL EXPERIMENTAL RESULTS

## D.1  DEFEND AGAINST BLACK-BOX ATTACK

In Section 5, we validate that our defense framework achieves a favorable trade-off between safety and utility against white-box attacks. In this section, we further evaluate its effectiveness against black-box attacks. Similarly, we use the official implementation of Poison-splat (Lu et al., 2024a) to simulate attacker behavior in black-box settings and the official implementations of Scaffold-GS (Lu et al., 2024b) to simulate victim behavior. As reported in Tables 6 and 7, our defense framework consistently outperforms naive defenses, yielding improvements in utility by more than 1 dB while maintaining comparable safety. The results indicate that our proposed framework demonstrates significant effectiveness in different settings and provides comprehensive safeguarding for real-world 3DGS systems.

## D.2  DEFEND AGAINST ATTACKS OF DIFFERENT STRENGTHS

We evaluate our defense framework under attacks of different strengths in the white-box setting. For the detector, we consider $\epsilon = 50/255$, $\epsilon = 100/255$, and $\epsilon = 150/255$, and report accuracy, F1 score, and recall metrics on mixed poisoned and clean datasets. As shown in Table 8, the detector consistently achieves a strong discrimination capability, demonstrating notable robustness against

Table 8. Performance of the detector in discriminating clean data from poisoned data across varying attack strengths on the Mip-NeRF360 dataset. The detector in our RemedyGS accurately distinguishes poisoned data of different attack strengths.

| Scene | Metric | Setting ($\epsilon$) | | |
|-------|--------|---------|---------|---------|
| | | 50/255 | 100/255 | 150/255 |
| MIP-bonsai | Accuracy | 0.9919 | 0.9939 | 0.9959 |
| | F1 | 0.9919 | 0.9939 | 0.9959 |
| | Recall | 0.9919 | 0.9939 | 0.9959 |
| MIP-all | Accuracy | 0.9905 | 0.9929 | 0.9962 |
| | F1 | 0.9905 | 0.9929 | 0.9962 |
| | Recall | 0.9905 | 0.9929 | 0.9962 |

Table 9. Comparison of baselines: *image smoothing* and *limiting the number of Gaussians* with the purifier in our RemedyGS framework on poisoned data under varying and stronger attack strengths. The evaluation is conducted on the Mip-NeRF360 dataset using utility metrics. Our RemedyGS demonstrates superior capability in recovering the original clean data from the poisoned data, even under stronger attack scenarios.

| Metric | PSNR ↑ | | | LPIPS ↓ | | | SSIM ↑ | | |
|--------|--------|--------|--------|--------|--------|--------|--------|--------|--------|
| Setting / Scene | image smoothing | limiting Gaussian number | RemedyGS (Ours) | image smoothing | limiting Gaussian number | RemedyGS (Ours) | image smoothing | limiting Gaussian number | RemedyGS (Ours) |
| Constrained Poison-splat attack with $\epsilon = 26/255$ | | | | | | | | | |
| MIP-bonsai | 28.398 | 22.680 | **30.212** | 0.287 | 0.476 | **0.254** | 0.832 | 0.656 | **0.891** |
| MIP-counter | 27.069 | 20.064 | **27.952** | 0.300 | 0.509 | **0.264** | 0.808 | 0.577 | **0.853** |
| MIP-kitchen | 28.641 | 20.402 | **29.088** | 0.231 | 0.446 | **0.186** | 0.844 | 0.593 | **0.871** |
| MIP-room | 27.963 | 21.221 | **29.662** | 0.331 | 0.486 | **0.283** | 0.785 | 0.563 | **0.855** |
| Constrained Poison-splat attack with $\epsilon = 35/255$ | | | | | | | | | |
| MIP-bonsai | 26.750 | 21.706 | **29.574** | 0.340 | 0.498 | **0.278** | 0.778 | 0.597 | **0.881** |
| MIP-counter | 25.709 | 19.112 | **27.296** | 0.340 | 0.533 | **0.283** | 0.760 | 0.533 | **0.832** |
| MIP-kitchen | 27.544 | 19.637 | **28.388** | 0.263 | 0.477 | **0.212** | 0.812 | 0.532 | **0.849** |
| MIP-room | 26.005 | 20.129 | **29.045** | 0.375 | 0.506 | **0.304** | 0.719 | 0.507 | **0.839** |
| Constrained Poison-splat attack with $\epsilon = 40/255$ | | | | | | | | | |
| MIP-bonsai | 25.739 | 20.830 | **28.871** | 0.367 | 0.509 | **0.291** | 0.742 | 0.569 | **0.859** |
| MIP-counter | 24.929 | 18.633 | **26.764** | 0.361 | 0.542 | **0.294** | 0.731 | 0.514 | **0.813** |
| MIP-kitchen | 26.910 | 19.105 | **27.664** | 0.282 | 0.490 | **0.227** | 0.792 | 0.505 | **0.831** |
| MIP-room | 25.078 | 19.751 | **28.517** | 0.402 | 0.517 | **0.313** | 0.676 | 0.480 | **0.821** |

different settings. We then assess the purifier with $\epsilon = 26/255$, $\epsilon = 35/255$, and $\epsilon = 40/255$, as detailed in Table 9 and Table 10. Baselines such as image smoothing and Gaussian limiting demonstrate weaker robustness and incur rapidly increasing computational cost as attack strengths are intensified, while our method remains effective even under stronger adversaries. In terms of utility, our approach also delivers superior 3D reconstruction. With increasing $\epsilon$, naive defenses degrade performance by up to 3 dB, whereas our method suffers far less degradation and consistently outperforms baselines. Overall, our framework achieves a more favorable trade-off between safety and utility across different attack strengths.

### D.2.1 IMPACT OF UNDETECTED POISONED IMAGES

In addition to investigating the impact of misclassified clean images on the system's utility performance in Table 4, we further investigate the effect of undetected poisoned images on system safety. Specifically, we assess how varying proportions of poisoned images that remain undetected and unpurified influence the system's safety performance. We combine 1%, 5%, 10% poisoned images with the remaining purified poisoned images in a scene and test the safety performance of the 3DGS system. The results are detailed in Table 11. The computational cost remains nearly identical to that observed for clean images across different mixed ratios, indicating that the presence of up to 10% poisoned images undetected in a scene almost has negligible influence on system safety. Furthermore, our detector demonstrates discriminative capability, achieving nearly 100% accuracy in identifying poisoned images. Even in cases where a very small fraction of poisoned images are misclassified and remain undetected, such errors have minimal to no effect on the overall safety of the system.

Table 10. Comparison of baselines: *image smoothing* and *limiting the number of Gaussians* with the purifier in our RemedyGS framework on poisoned data under varying and stronger attack strengths. The evaluation is conducted in terms of computation cost metrics on Mip-NeRF360. Our RemedyGS exhibits the strongest capability in defending against even more intensive computation cost attacks.

| Metric | Number of Gaussians | | | Peak GPU memory [MB] | | | Training time [minutes] | | | Rendering speed [FPS] | | |
|---|---|---|---|---|---|---|---|---|---|---|---|---|
| Setting
Scene | image
smoothing | limiting
Gaussian
number | RemedyGS
(Ours) | image
smoothing | limiting
Gaussian
number | RemedyGS
(Ours) | image
smoothing | limiting
Gaussian
number | ours | image
smoothing | limiting
Gaussian
number | RemedyGS
(Ours) |
| Constrained Poison-splat attack with $\epsilon = 26/255$ | | | | | | | | | | | | |
| MIP-bonsai | 0.798 M | 2.148 M | 1.031 M | 8912 | 11590 | 8731 | 17.04 | 19.58 | 17.77 | 307 | 168 | 260 |
| MIP-counter | 0.772 M | 2.099 M | 0.966 M | 8693 | 10074 | 9133 | 19.92 | 20.38 | 20.79 | 247 | 194 | 201 |
| MIP-kitchen | 1.017 M | 2.101 M | 1.515 M | 9861 | 10418 | 10521 | 22.79 | 21.97 | 23.08 | 199 | 146 | 149 |
| MIP-room | 0.961 M | 2.150 M | 1.102 M | 11120 | 12185 | 10374 | 20.33 | 20.10 | 20.97 | 213 | 153 | 182 |
| Constrained Poison-splat attack with $\epsilon = 35/255$ | | | | | | | | | | | | |
| MIP-bonsai | 0.972 M | 2.169 M | 0.980 M | 8659 | 10814 | 8603 | 17.80 | 20.03 | 17.59 | 272 | 163 | 264 |
| MIP-counter | 0.832 M | 2.117 M | 0.917 M | 9190 | 9785 | 8870 | 20.55 | 20.00 | 20.26 | 218 | 204 | 215 |
| MIP-kitchen | 1.144 M | 2.130 M | 1.533 M | 10056 | 10582 | 10340 | 23.44 | 22.44 | 23.14 | 180 | 146 | 148 |
| MIP-room | 1.289 M | 2.167 M | 1.015 M | 13269 | 12652 | 10282 | 22.24 | 20.23 | 20.45 | 161 | 147 | 201 |
| Constrained Poison-splat attack with $\epsilon = 40/255$ | | | | | | | | | | | | |
| MIP-bonsai | 1.130 M | 2.186 M | 0.948 M | 8850 | 11158 | 8586 | 18.28 | 18.19 | 17.56 | 137 | 169 | 261 |
| MIP-counter | 0.894 M | 2.122 M | 0.909 M | 9548 | 9885 | 9237 | 20.72 | 20.15 | 17.43 | 207 | 200 | 205 |
| MIP-kitchen | 1.313 M | 2.132 M | 1.692 M | 10508 | 10532 | 10790 | 24.09 | 20.15 | 22.26 | 229 | 148 | 137 |
| MIP-room | 1.625 M | 2.159 M | 0.986 M | 15710 | 12514 | 10227 | 23.31 | 18.89 | 20.44 | 174 | 149 | 208 |

Table 11. Impact of different ratios of undetected poisoned data in a scene on safety performance in term of computation cost metrics including the number of Gaussians, peak GPU memory, training time, and rendering speed.

| Metric | Number of Gaussians | | | Peak GPU memory [MB] | | | Training time [minutes] | | | Rendering speed [FPS] | | |
|---|---|---|---|---|---|---|---|---|---|---|---|---|
| Setting
Scene | clean | attack | Mixed | clean | attack | Mixed | clean | attack | Mixed | clean | attack | Mixed |
| Mixed ratio: 1% | | | | | | | | | | | | |
| NS-chair | 0.495 M | 0.942 M | 0.492 M | 4633 | 9378 | 4834 | 7.81 | 12.41 | 9.63 | 345 | 170 | 354 |
| NS-ficus | 0.267 M | 0.288 M | 0.220 M | 4052 | 5447 | 4057 | 6.41 | 10.90 | 8.66 | 606 | 447 | 647 |
| MIP-room | 1.536 M | 7.368 M | 1.177 M | 11370 | 47721 | 10879 | 22.14 | 48.11 | 20.71 | 154 | 31 | 191 |
| Mixed ratio: 5% | | | | | | | | | | | | |
| NS-chair | 0.495 M | 0.942 M | 0.503 M | 4633 | 9378 | 4922 | 7.81 | 12.41 | 8.69 | 345 | 170 | 338 |
| NS-ficus | 0.267 M | 0.288 M | 0.220 M | 4052 | 5447 | 4050 | 6.41 | 10.90 | 8.02 | 606 | 447 | 615 |
| MIP-room | 1.536 M | 7.368 M | 1.178 M | 11370 | 47721 | 10954 | 22.14 | 48.11 | 20.67 | 154 | 31 | 199 |
| Mixed ratio: 10% | | | | | | | | | | | | |
| NS-chair | 0.495 M | 0.942 M | 0.520 M | 4633 | 9378 | 4906 | 7.81 | 12.41 | 9.79 | 345 | 170 | 314 |
| NS-ficus | 0.267 M | 0.288 M | 0.223 M | 4052 | 5447 | 4060 | 6.41 | 10.90 | 8.89 | 606 | 447 | 616 |
| MIP-room | 1.536 M | 7.368 M | 1.165 M | 11370 | 47721 | 10977 | 22.14 | 48.11 | 20.47 | 154 | 31 | 196 |

## D.3 DEFENSE PERFORMANCE IN TERMS OF TRAINING TIME AND RENDERING SPEED ACROSS MULTIPLE DATASETS.

We provide additional evaluation results on the computational cost in terms of training time and rendering speed, as an extension of Table 1. As shown in Table 12, the results indicate that our method mitigates the increase in computational overhead, achieving the same level of computational cost comparable to that of clean images.

## D.4 ADDITIONAL VISUALIZATION RESULTS

We provide additional qualitative results of our method in Figure 5. The results demonstrate that although limiting the number of Gaussians imposes an upper bound on the computational cost, sharpened regions force reallocation of Gaussians from unsharpened areas, leaving certain regions without Gaussian representation and consequently diminishing local details. In contrast, our method preserves nearly the same level of detail as the clean images.

Additional visualizations of the recovered images by different methods are shown in Figure 6. It confirms that adversarial training effectively alleviates local over-smoothing issues, thereby enhancing perceptual fidelity and enabling higher-quality 3D representations.

Table 12. Comparison of baselines: *image smoothing* and *limiting the number of Gaussians* with our RemedyGS framework on poisoned data, evaluated in terms of computational cost metrics including training time and rendering speed across NeRF-Synthetic, Mip-NeRF360 and Tanks-and-Temples datasets. Our RemedyGS effectively mitigates the negative impact of computation cost attacks.

| Metric | Training time [minutes] | | | | | Rendering speed [FPS] | | | | |
|---|---|---|---|---|---|---|---|---|---|---|
| Setting
Scene | clean | poisoned | image smoothing | limiting Gaussian number | ours | clean | poisoned | image smoothing | limiting Gaussian number | ours |
| NS-chair | 7.81 | 12.41 | 9.71 | 8.61 | 10.07 | 345 | 170 | 645 | 371 | 348 |
| NS-ficus | 6.41 | 10.90 | 9.08 | 9.16 | 9.48 | 606 | 447 | 797 | 471 | 701 |
| NS-hotdog | 7.68 | 16.23 | 9.31 | 9.98 | 10.85 | 655 | 100 | 852 | 356 | 374 |
| NS-lego | 7.36 | 13.77 | 9.96 | 9.52 | 10.34 | 483 | 194 | 702 | 403 | 445 |
| NS-Avg | 7.30 | 12.87 | 9.66 | 9.61 | 10.17 | 566 | 236 | 786 | 373 | 489 |
| MIP-bonsai | 16.66 | 33.72 | 13.98 | 18.11 | 18.63 | 222 | 60 | 290 | 178 | 277 |
| MIP-counter | 22.21 | 35.46 | 17.79 | 20.67 | 23.7 | 175 | 55 | 240 | 192 | 213 |
| MIP-kitchen | 22.84 | 41.99 | 19.56 | 20.05 | 23.64 | 138 | 46 | 205 | 157 | 168 |
| MIP-room | 22.14 | 48.11 | 17.26 | 19.45 | 23.86 | 154 | 31 | 225 | 147 | 185 |
| MIP-Avg | 25.37 | 40.62 | 18.94 | 24.94 | 24.59 | 128 | 51 | 197 | 129 | 157 |
| TT-Barn | 13.54 | 17.03 | 12.19 | 10.52 | 11.61 | 265 | 135 | 481 | 388 | 343 |
| TT-Francis | 10.19 | 13.47 | 10.47 | 9.71 | 10.15 | 300 | 159 | 497 | 349 | 477 |
| TT-M60 | 13.85 | 19.32 | 11.68 | 14.45 | 10.88 | 188 | 103 | 331 | 194 | 301 |
| TT-Panther | 14.68 | 20.58 | 9.63 | 14.56 | 10.07 | 179 | 95 | 390 | 205 | 309 |
| TT-Avg | 15.27 | 19.60 | 11.71 | 14.63 | 12.59 | 194 | 122 | 347 | 231 | 288 |

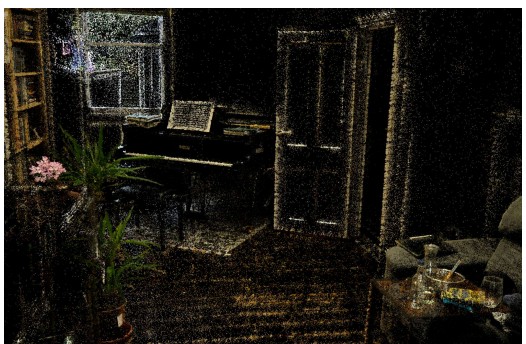

**Rendering with Clean Image Input**

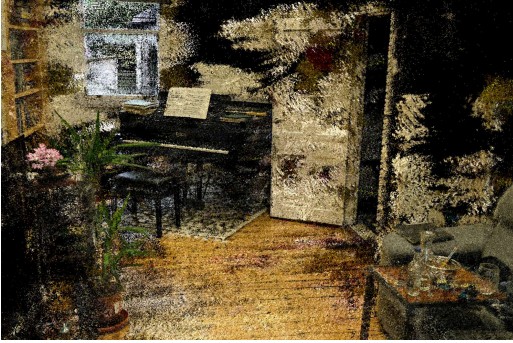

**Rendering with Attacked Image Input**

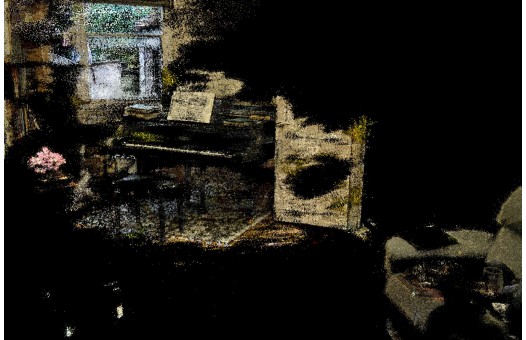

**Rendering with Limited Gaussian Number**  **Rendering under RemedyGS(Ours)**

Figure 5. 3D Gaussian point cloud visualization of rendering results from different input images. Top row: (Left) Point cloud visualization of rendering from clean image input. (Right) Point cloud visualization of rendering from attacked image input. Bottom row: (Left) Point cloud visualization of rendering from the limiting Gaussian number defense method. (Right) Point cloud visualization of rendering from our RemedyGS method.

## E  THE USAGE OF LARGE LANGUANGE MODELS (LLMS)

During the preparation of this manuscript, large language models (LLMs) were utilized solely as writing assistants to refine language expression, including improvements in clarity, grammar, and

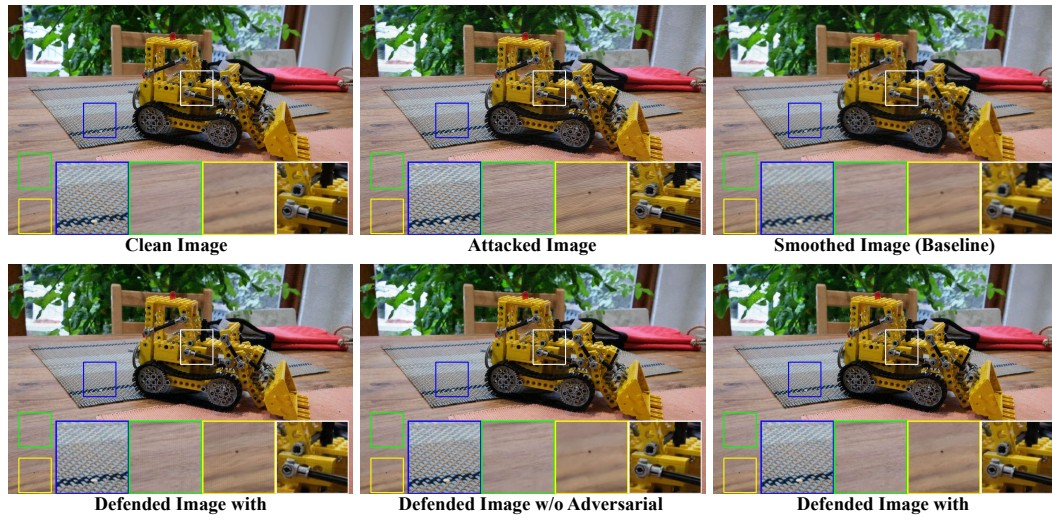

**Clean Image**       **Attacked Image**       **Smoothed Image (Baseline)**

**Defended Image with**    **Defended Image w/o Adversarial**    **Defended Image with**
**Concatenated Purifier**            **Training**             **RemedyGS (Ours)**

Figure 6. Qualitative results for the kitchen scene in the Mip-NeRF360 dataset. Top row: (left) clean input image, (middle) attacked input image, (right) defended input image using the image smoothing baseline. Bottom row: (left) purified input image using the purifier with concatenated skip connections, (middle) purified input image using the purifier with added skip connections but without adversarial training, (right) purified input image using the purifier in our RemedyGS framework. Our method achieves the best recovery performance among all baselines and effectively mitigates the issue of local over-smoothing compared with methods without adversarial training.

readability. All substantive intellectual contributions, encompassing the conception of research ideas, the development of methodologies, and the design of experiments, were carried out exclusively by the authors without the involvement of LLMs.

