# OpenReview forum: "RemedyGS: Defend 3D Gaussian Splatting Against Computation Cost Attack"
_ICLR.cc/2026/Conference — ICLR 2026 Conference Withdrawn Submission_

### Official Review · Reviewer_9GDn · 2025-10-28

**Soundness:** 2
**Presentation:** 2
**Contribution:** 2
**Rating:** 4
**Confidence:** 3

**Summary:**

The paper proposes RemedyGS, a defense pipeline for computation-cost (DoS-style) attacks on 3D Gaussian Splatting (3DGS). The system first uses a detector to flag poisoned inputs that exhibit high-frequency/TV artifacts, then applies a learnable purifier (encoder–decoder with additive skip connections) to recover a “clean” image; a discriminator is added for adversarial training to improve perceptual quality. Experiments against Poison-splat show sizable reductions in Gaussian count and GPU memory while retaining reconstruction fidelity, often outperforming two baselines (global smoothing; hard caps on Gaussian count).

**Strengths:**

- The two-stage “detect then purify” design is simple and service-friendly (avoid touching benign inputs), and the adversarial fine-tuning motivation is reasonable.
- The Barber–Agakov lower-bound derivation justifies an MSE-style training target to maximize mutual information between purified and original images. It’s neat and easy to implement.
- On NeRF-Synthetic/Mip-NeRF360/Tanks&Temples, RemedyGS generally restores PSNR/SSIM and lowers memory/Gaussian count close to clean runs, outperforming smoothing and hard-caps.
- Results under stronger ε constraints indicate the approach scales more gracefully than baselines.

**Weaknesses:**

- The core recipe—CNN detector + image-to-image purifier + GAN-style discriminator—feels standard for “detect-then-denoise” pipelines; transplanting it to 3DGS is useful but not conceptually deep. The paper’s technical lift is modest beyond the BA bound restatement and some architectural choices.
- The method is positioned as a black-box defense against a white-box attack, but there is no evaluation against adaptive attackers that (i) craft poisons to evade the detector, or (ii) directly attack the purifier/discriminator (e.g., gradient-free queries to break LPIPS/MSE alignment). This leaves open whether RemedyGS remains robust once the attacker targets the detector’s features or adversarially optimizes to fool the purifier.
- The detector is trained on paired clean/poison data (DL3DV) and tested on other datasets; success is shown, but the distribution shift story and decision thresholds are not deeply analyzed (false-positive/false-negative costs in a live service, robustness to different cameras/textures).
- The skip-connection and adversarial components are ablated, but there’s no study of alternative detectors (frequency/TV-feature models, ViT-based), no systematic threshold sweeps, and no reporting of ROC/PR curves under varying ε to characterize detectability limits.

**Questions:**

See weakness as above.

---

### Official Review · Reviewer_kxhu · 2025-10-30

**Soundness:** 3
**Presentation:** 3
**Contribution:** 3
**Rating:** 6
**Confidence:** 2

**Summary:**

The authors present RemedyGS, which proposes a two-stage, black-box defense for 3D Gaussian Splatting (3DGS) DoS-style “computation cost” attacks (e.g., Poison-splat): a detector flags poisoned inputs and a purifier restores them; a GAN-style adversarial training step further aligns purified outputs with the clean-image distribution to preserve fidelity. Compared with naive defenses (global smoothing or capping Gaussian count), RemedyGS brings computational cost near benign levels while maintaining better PSNR/SSIM/LPIPS across NeRF-Synthetic, Mip-NeRF360, and T&T .

**Strengths:**

First comprehensive black-box defense tailored to 3DGS computation-cost attacks; system-agnostic pipeline with detector + purifier.

Adversarially trained purifier improves perceptual quality, mitigating over-smoothing versus plain MSE (ablation shows LPIPS/PSNR gains).

High detector accuracy across datasets and clear DoS threat framing with Poison-splat baseline setup.

**Weaknesses:**

Black-box evaluation details deferred: main text focuses on white-box; black-box defense results are pushed to Appendix, limiting immediate scrutiny of generalization to unseen attackers.

Baselines are limited to image smoothing and Gaussian-count capping; no comparison to more advanced, learned purifiers/detectors beyond the proposed one.

Potential residual blur trade-offs remain (the purifier needed adversarial training to counter MSE smoothing), suggesting sensitivity to training choices and loss weights.

Lack of adaptive attack discussions and evaluations.

**Questions:**

See the weakness section. More baselines and discussion around robustness to adaptive attacks are needed.

---

### Official Review · Reviewer_yoRA · 2025-10-30

**Soundness:** 3
**Presentation:** 3
**Contribution:** 2
**Rating:** 2
**Confidence:** 3

**Summary:**

This paper addresses a vulnerability in 3DGS systems where adversarial perturbations added to training images can cause denial-of-service attacks by triggering excessive Gaussian primitive generation during reconstruction. The authors propose RemedyGS, a defense framework consisting of two main components: a detector and a purifier. The detector is trained using adversarial training to distinguish between clean and poisoned images, while the purifier employs a variational information bottleneck approach to filter out attack-induced perturbations while preserving scene content. The framework is designed to be system-agnostic, operating as a preprocessing module before existing 3DGS reconstruction pipelines. Experimental evaluation demonstrates that the defense can effectively detect poisoned images and restore reconstruction quality across multiple benchmark datasets while maintaining quality for benign users.

**Strengths:**

S1: The paper tackles a genuine security vulnerability in 3DGS systems with practical implications for commercial services. Denial-of-service attacks through adversarial image poisoning represent a realistic threat to cloud-based platforms. The problem formulation is well-motivated with clear attack scenarios and potential consequences.

S2: The proposed defense adopts a practical modular design that integrates with existing 3DGS systems without requiring core algorithm modifications. The two-stage detector-purifier architecture provides interpretability by separating detection and remediation functions. The system-agnostic approach allows deployment across different 3DGS variants.

S3: The paper provides ablation studies examining key design choices and analyzes performance under varying attack strengths. Qualitative visualizations effectively complement quantitative metrics in demonstrating effectiveness.

**Weaknesses:**

W1: The paper's threat model is fundamentally flawed and overly restrictive. The defense is designed exclusively against Poison-splat, a white-box attack that requires attackers to have full access to the 3DGS training pipeline and perform exhaustive per-scene optimizations. In real-world scenarios, attackers are unlikely to possess such comprehensive knowledge of the victim's system architecture, hyperparameters, and training procedures.

W2: The most critical flaw is the complete lack of adaptive attack evaluation. Security defenses must be tested against adversaries who know the defense exists and actively try to bypass it. The paper only evaluates against oblivious attacks where the attacker is unaware of RemedyGS, which is an unrealistic assumption once the defense is publicly deployed. An attacker could adversarially train against the detector to learn evasion patterns or exploit gradient masking techniques. The purifier is trained on a fixed attack distribution from Poison-splat's specific optimization procedure, making it vulnerable to attacks with different statistical properties or alternative generation mechanisms. Attackers could also design perturbations that remain effective even after purification by incorporating the purifier into their attack optimization loop.

W3: Image resolution handling is inconsistent and unexplained: the original DL3DV dataset uses 960×540, the detector is trained on 960×528, the purifier on 960×544, and evaluation uses yet different resolutions such as 1600×1066 for Mip-NeRF360. These arbitrary resolution changes cause aspect ratio distortions that could significantly impact performance, yet no explanation or ablation is provided.

W4: The paper provides no evaluation of the computational overhead introduced by the detector and purifier inference, which could negate the benefits of preventing DoS attacks.

W5: Only two simplistic baseline defenses are compared: Gaussian filtering and Gaussian number limiting. Are there any advanced defense techniques that can be compared, such as adversarial example detectors, backdoor defense mechanisms, and robust training methods?

W6: Training uses only 320 scenes from DL3DV, which is extremely limited compared to real-world 3DGS application diversity. More critically, training requires paired clean and poisoned images. How does one obtain large-scale poisoned training data before attacks occur in practice? If attacks are not yet widespread, where do training samples come from? If relying on simulated attacks, performance against real attacks cannot be guaranteed.

W7: The paper does not report inference time, memory footprint, or energy consumption of the defense components, nor does it analyze scalability across different hardware configurations. For real-time or near-real-time 3DGS services, the overhead analysis is important.

W8: Although Table 3 reports F1 and Recall, the False Positive Rate (FPR) remains unreported. In real-world deployment scenarios where benign users constitute the vast majority, FPR more directly reflects the impact on normal users than Precision does. Table 4 only displays reconstruction metrics for 4 scenes without clarifying how many clean images were misclassified or showing the quality degradation of these false positives after purification.

**Questions:**

Q1: Tables 1 and 2 do not specify the attack strength parameter ε used for these results. Which ε value was used for the main evaluation?

Q2: The loss weights α1=0.23, α2=100, α3=1 seem unusual, with LPIPS weighted 100× higher than MSE. What is the rationale behind this weighting scheme?

---

### Official Review · Reviewer_ToPY · 2025-10-30

**Soundness:** 2
**Presentation:** 2
**Contribution:** 2
**Rating:** 2
**Confidence:** 5

**Summary:**

- The paper proposes RemedyGS, a two-stage black-box defense for 3D Gaussian Splatting (3DGS) against computation-cost (DoS-style) attacks such as Poison-splat. It detects poisoned inputs via a lightweight CNN classifier trained on texture cues, and purifies detected images with an encoder–decoder trained with MSE + LPIPS, further refined with adversarial training via a discriminator conditioned on latent features. Across NeRF-Synthetic, Mip-NeRF360, and Tanks-and-Temples, RemedyGS claims to bring Gaussian counts / GPU memory close to benign runs and substantially recover PSNR/SSIM/LPIPS relative to simple baselines (smoothing, bounding Gaussian count).

**Strengths:**

- The paper tackles a real and novel security angle for 3DGS with a pragmatic, easy-to-deploy pipeline and strong reported gains over simple baselines.

**Weaknesses:**

- The defense is only tested against the original Poison-splat setup, without considering stronger adaptive variants. For example, it would be valuable to test (a) a constrained, differentiable attacker that explicitly attempts to bypass the detector by minimizing total variation while still maximizing densification, and (b) a purifier-aware attack that optimizes under the expectation of the purification process to preserve the malicious signal after denoising.
- RemedyGS’s purification stage is based on an encoder–decoder with skip connections and adversarial refinement, yet the paper does not compare it against established image-restoration models such as UNet or DnCNN trained on the same poisoned–clean pairs. Moreover, recent diffusion-based restorers (e.g., Imagen or Stable Diffusion variants fine-tuned for artifact removal) represent a relevant and much stronger baseline. Without such comparisons, it’s difficult to judge whether RemedyGS’s advantage arises from its architectural design or simply from the dataset and training setup.
- While the paper discusses improvements in Gaussian count, training time, and GPU memory after applying the defense, it does not isolate or report the additional cost introduced by the detector or purifier themselves. There are no runtime measurements (e.g., per-image latency, throughput reduction, or added memory usage) that would allow assessing the practicality of RemedyGS in large-scale or real-time 3DGS deployments.

**Questions:**

Please check weaknesses.

---

### Note · Authors · 2025-11-13

I have read and agree with the venue's withdrawal policy on behalf of myself and my co-authors.